# Canine Prostate Cancer: Current Treatments and the Role of Interventional Oncology

**DOI:** 10.3390/vetsci11040169

**Published:** 2024-04-09

**Authors:** Erin A. Gibson, William T. N. Culp

**Affiliations:** 1Department of Clinical Sciences and Advanced Medicine, School of Veterinary Medicine, University of Pennsylvania, Philadelphia, PA 19123, USA; 2Department of Veterinary Surgical and Radiological Sciences, School of Veterinary Medicine, University of California-Davis, Davis, CA 95616, USA

**Keywords:** stent, embolization, radiotherapy, prostatectomy, interventional radiology

## Abstract

**Simple Summary:**

Prostate carcinoma remains a therapeutic challenge in veterinary medicine. Current treatment focuses on locoregional control ideally while minimizing morbidity, as well as systemic therapy for the management of distant disease progression. Below, the current treatment modalities, including the role of interventional oncology in the management of prostate carcinoma therapy in dogs, are reviewed. Additionally, the role of dogs as a translational model for research in people is acknowledged, as well as the consideration of using therapeutic strategies commonly utilized for people for dogs.

**Abstract:**

Prostate carcinoma is one of the most common cancers worldwide in men, with over 3 million men currently living with prostate carcinoma. In men, routine screening and successful treatment schemes, including radiation, prostatectomy, or hormone therapy, have allowed for high survivability. Dogs are recognized as one of the only mammals to spontaneously develop prostate neoplasia and are an important translational model. Within veterinary medicine, treatment options have historically been limited in efficacy or paired with high morbidity. Recently, less invasive treatment modalities have been investigated in dogs and people and demonstrated promise. Below, current treatment options available in dogs and people are reviewed, as well as a discussion of current and future trends within interventional treatment for canine PC.

## 1. Introduction

Most tumors arising from the prostate are histopathologically characterized as prostate adenocarcinoma or urothelial carcinomas (UC), although other types have been reported [1]. Prostate carcinoma (PC) may arise from acinar epithelium urothelium lining the prostatic urethra or ductal epithelium, and it remains challenging to distinguish prostate-origin carcinoma from urothelial carcinoma arising from the urethral or prostatic ducts and invading the prostate secondarily. Histopathologically, both UC and PC demonstrate a heterogenous appearance, and histologic differences do not appear to be correlated with clinical outcomes in dogs [1,2,3]. Additional histopathological investigation in canine PC and hyperplasia suggested that different cell populations are susceptible to neoplastic transformation (ductal cells) compared to hyperplastic age-related steroid-responsive change (basal cells). This finding is also supportive of the noticeable risk of castrated dogs for PC compared to intact male dogs [4,5]. 

Most PCs are diagnosed in older castrated male dogs [2,3,4,5]. In general, PC is not identified until later stages of the disease when clinical signs such as dysuria, hematuria, dyschezia, hind limb pain, or ataxia are noted with urogenital signs preceding gastrointestinal and systemic signs typically [6,7]. Metastatic disease is most frequently diagnosed in the lung, lumbar spine/pelvis, or lumbar lymph nodes [2,3]. Pulmonary metastatic rate at the time of diagnosis ranges from 8 to 50% [5,8], while local metastasis, such as lymph node and bone, ranges from 15 to 72% [5,9], and gross metastasis is reportedly >80% at the time of death [3,5]. 

## 2. Diagnosis

The diagnosis of PC in men often consists of chemical marker assay screening such as prostate-specific antigen (PSA), as well as prostate biopsy [10]. Currently, there are no such screening tools able to identify benign or malignant diseases of the canine prostate or distinguish between malignant prostatic or urothelial cell origin, which creates an inherent challenge to diagnosing canine PC [1]. Recently, BRAF gene mutations have been discovered in a majority of canine PC and UC [11], which are associated with pro-oncogenic properties and can also be detected in urine samples in dogs with BRAF mutation containing UC or PC [12]. Imaging, including ultrasound, computed tomography (CT), or radiographs, may be performed. Changes such as mineralization, regional lymphadenopathy, loss of parenchymal architecture, and prostate capsule integrity are consistent with canine PC. The mineralization of the prostate gland in neutered dogs is strongly associated with neoplasia; however, this finding in intact dogs is less conclusive [13]. Radiographs or CTs may identify bone lesions consistent with distant metastasis or mineralization in the region of the prostate [2,14]. Cytologic diagnosis acquired by ultrasound-guided fine-needle aspiration, diagnostic catheterization, or urine sediment cytology are reported. There appears to be a strong correlation between cytologic diagnosis and histopathologic diagnosis [15], although the manner by which cytology is acquired is important. Diagnostic catheterization appears to be highly sensitive and specific for UC/PC cytology, and utilization of pathologist review may help improve the sensitivity and specificity of this diagnostic [16]. Importantly, the seeding of the abdominal wall following fine-needle aspiration or percutaneous biopsy of UC/PC was described [17,18], and caution should be used when considering this diagnostic method. 

## 3. Current Treatments

### 3.1. Medical Management

The medical treatment of PC in dogs includes the use of non-steroidal anti-inflammatory drugs and chemotherapy. The expression of cyclooxygenases (Cox)-1 and -2 was evaluated in PC in dogs. Cox-1 was detected in normal and neoplastic prostatic epithelial cells, while Cox-2 was exclusively identified in tumor cells, and both were identified in the majority of tumors (94% and 88%, respectively) for which they were evaluated. While Cox-1 and -2 positive tumors do not appear to have significantly different clinical courses compared to negative tumors, it does justify the use of NSAIDS in these patients; the clear superiority of one NSAID over another has not been established [9]. Importantly, the anti-tumor effects of Cox inhibitors are likely multifactorial, although they may act on Cox-dependent or -independent pathways [9]. The median survival time (MST) in dogs receiving NSAIDS vs. no treatment was 6.9 mo compared to 0.7 mo, which was significant [9]. In contrast to NSAIDS, the antitumor effects of chemotherapy in the treatment of PC appears to be generally poor. A retrospective study evaluating mitoxantrone paired with piroxicam in dogs with PC objectively identified no partial or complete response, although most owners perceived improvement in urination and/or defecation. In this population, MST for all dogs was 155 days [8]. A more recent prospective open-label phase III randomized study compared mitoxantrone to carboplatin administered every 3 weeks with piroxicam concurrently in dogs with lower urinary tract tumors, including PC. There was no significant difference in treatment arms, but similarly, prostatic involvement appeared to negatively impact survival with a median survival of 109 days compared to urethral, trigonal, or apically located tumors (300, 190, and 645 days, respectively) [19]. The authors suggest that the addition of chemotherapy may prolong survival in dogs based on their results compared to historical published data, despite not having a piroxicam-only treatment arm. A similar suggestion was made following retrospective evaluation of PC treated with NSAIDS with or without chemotherapy, which found that dogs treated with NSAIDS and chemotherapy had significantly longer MST and time to progression (106 d, 76 d, respectively) compared to NSAIDS alone (51 d and 44 d, respectively) [6]. While this is compelling, there is a lack of prospective randomized controlled trials (RCT) dictating the difference in outcomes between NSAIDS alone and in conjunction with chemotherapy, which softens the recommendation to apply the two concurrently. In patients for which chemotherapy is available and is likely to be well tolerated, concurrent application can be considered. 

### 3.2. Surgery

In people, definitive intent treatment options for PC in men include surgery or radiotherapy, with surgery possibly being preferred over radiotherapy for impact on overall and PC-specific mortality in patients [20]; differences in bowel and genitourinary symptoms may be inconsistent between the two, however [21]. In dogs, prostatectomy was evaluated. A study comparing dogs with PC treated with NSAIDS with or without chemotherapy (e.g., toceranib, carboplatin, 5-fluorouracil, and chlorambucil) had a median survival time of 90 days following diagnosis. This was compared to dogs who underwent surgical treatment {total prostatectomy (TP) or total prostatocystectomy (TPC)} with a median survival time of 337 days; dogs survived significantly longer in the TP group (>500 days) compared to the TPC group (83 d). In that study, most patients (80%) experienced urinary incontinence following surgery [22]. In another report of TP with various reconstructive surgeries described in dogs with PC, MST was 231 days, and permanent incontinence was reported in one-third of dogs [23]. While surgery is a viable option, it remains moderately morbid and reasonably complicated, with a high risk of urinary incontinence. Therefore, the need for alternative loco-regional therapies is clear. 

### 3.3. Radiation

Curative intent external beam or brachytherapy was performed on men with high cure rates and mixed side effects compared to surgery, with some evidence suggesting improved urinary and sexual effects [24]. It was also used as an adjunct with prostatectomy [24]. Image-guided intensity-modulated radiotherapy (IMIG-RT) was described as a first-line or salvage procedure, with or without chemotherapy, in dogs with lower urinary tract carcinomas [7,25,26,27,28]. In studies evaluating the risk of acute radiation effects associated with pelvic irradiation with curative intent, external beam radiation gastrointestinal complications (specifically colitis) were encountered most commonly (38–75%) [26,27]. Studies similarly evaluating late complications for dogs receiving a definitive intent irradiation of pelvic region tumors identified one or more complications in 39–56% of patients, with necrotic drainage/ulceration of the skin and subcutaneous tissues within the radiation field, chronic colitis, strictures, and osteopenia being most commonly reported [26,27]. Interestingly, the perineal location was specifically identified as a riskier location for the development of complications, as was a larger radiation field. Therefore, irradiation in the region of the lower urinary tract is considered a lower risk in the scheme of pelvic RT [26]. A retrospective study evaluating the role of radiation therapy with or without concurrent chemotherapy in lower urinary tract carcinomas reported an event-free survival (EFS) of 260 days and an overall survival time (OST) of 510 days. All dogs were retrospectively categorized into three treatment groups, including dogs undergoing first-line concurrent chemoradiotherapy (1), first-line chemotherapy > 1 mo prior to initiating radiotherapy who did not have evidence of tumor progression (2), and dogs receiving radiotherapy as salvage following locoregional failure (3). Fifty-one dogs with primary genitourinary urothelial carcinoma were included and further categorized into bladder (19), prostate (17), and urethral (4), and eleven were multifocal within the urinary tract. Dogs with prostate involvement were not separately evaluated for factors such as acute or late side effects, although overall median survival times in dogs with prostate involvement was 341 days, which was significantly worse than for dogs without prostate involvement. Acute radiation effects were predominantly mild but were reported in 65% of treated dogs and included acute colitis most commonly, followed by acute dermatitis and genitourinary effects. Importantly, there was a 31% risk of permanent urinary incontinence, and late effects, including urethral stricture, were documented but uncommon. In all dogs, the median time to local progression was 343 days and was reported in 59% of dogs. Locoregional failure rates per group were 56%, 50%, and 75% for groups 1, 2, and 3 [25]. In a separate study that retrospectively evaluated the late effects of intensity-modulated image-guided radiotherapy (IMIG-RT) for genitourinary carcinomas in dogs, including PC and UC, late effects were only identified in 19% of dogs, which included grade 3 [29] genitourinary and gastrointestinal events. Acute effects occurred in the majority of patients with gastrointestinal (colitis) being most common, followed by integumentary and urinary tract effects. In this study, median EFS and OST was 317 and 654 days each, and the location of tumor did not appear to significantly affect outcome. Of the owners who completed standardized post-treatment questionaries, 60% perceived improved quality of life while 30% reported unchanged [28]. A more recent study solely evaluated definitive intent intensity-modulated radiation therapy for PC with or without concurrent chemotherapy. The median EFS and OST for all dogs were 220 and 563 days, respectively [7]. Within the treated population, the median time to local progression was 241 d, and 56% of patients had documentation of metastatic disease at a median of 108 days. In this population, the presence of symptoms at time of diagnosis negatively impacted survival, and EFS was shorter in patients with metastatic disease at diagnosis compared to those who did not have metastatic disease. Patients with the involvement of additional uroepithelia sites beyond the prostate did not have significantly different OST or EFS. Importantly, 60% of patients had grade 1–2 [29] acute toxicity documented while the estimated rate of late effects at 12 and 18 mo was 8% and 22% [7]. Metastatic disease was the most common reason for euthanasia, which suggests that aggressive local treatment should be paired with systemic treatment as well. Despite evidence demonstrating prolonged OST and EFS with systemic therapy, overall prognosis remains guarded with most animals succumbing to metastatic disease. 

## 4. Interventional Oncology Approaches to Prostate Carcinoma

Interventional oncology (IO) is the treatment of cancer using image-guided minimally invasive techniques. Available IO options include both definitive- and palliative-intent treatments. In veterinary medicine, IO techniques are particularly exciting due to the optimization of quality of life with a lowered morbidity. While still emerging in veterinary medicine, prospective and retrospective investigations of outcomes in dogs undergoing these treatments have started to guide the role of IO in the treatment of PC. 

### 4.1. Prostate Artery Embolization

Prostate artery embolization (PAE) is a minimally invasive technique involving the delivery of embolic material into the arterial blood supply that feeds the prostate. The prostate is a bilobed structure with independent blood supply per lobe. In most dogs, the internal pudendal artery branching from the internal iliac gives rise to the main prostatic artery. The prostatic artery also provides a smaller terminal branch, the caudal vesical artery, that courses towards the distal ureter and urethra and provides some supply. Distally, the prostate artery provides the small middle rectal artery as well as the three smaller terminal cranial, middle, and caudal prostate arteries [30]. To adequately embolize the prostate, the selection of the left and right prostate artery is attempted and, in all published descriptions, femoral or carotid artery access was elected. In people, PAE was investigated for the amelioration of lower urinary tract signs associated with benign prostate hyperplasia (BPH) in men due to its minimally invasive nature [31,32]. It was found to be technically safe, with good long term outcomes for the reduction of adverse symptoms and prostate volume [31,32]. Additionally, there has been some investigation into PAE for prostate bleeding associated with PC or for a tumoricidal effect in patients with localized PC. While technically successful in a majority of cases, there was evidence of the incomplete and non-sustainable control of PC [33,34]. The control of bleeding in men with advanced PC appears generally successful [35], although PAE is not considered a standard first line treatment for PC in men. Prior to translation in people, the embolization of canine prostates following the induction of BPH in a research setting was reported. In two early studies, PAE performed with microspheres in research dogs with induced BPH was found to be technically safe and feasible [36,37]. In a similar study evaluating the delivery of polyethylene glycol microspheres sized 400 +/− 75 μm in a spontaneous BPH model in intact beagles, a significant reduction in prostate volume was noted at 2- and 4-weeks post-embolization. A histopathological exam revealed diffuse glandular atrophy and interstitial fibrosis, although the partial or complete recanalization of all prostate arteries was demonstrated 1 mo following initial embolization [38]. 

Prostate artery embolization with embolic beads in dogs with spontaneous PC was also evaluated. It was found to be technically successful, and all dogs had a reduction in prostate volume after PAE with a median decrease in prostate volume of 39.4% as measured on CT 1 mo following treatment. Additionally, the clinical signs of stranguria, tenesmus, and lethargy were significantly less common 30 days after PAE compared to before [14]. Drug-loaded beads (DEB) with docetaxel were investigated for use in a canine model of spontaneous prostate carcinoma and evaluated at day 30 and day 60 following embolization [39]. Three of five dogs were unable to make it to the endpoint of study due to rapid disease progression, although CT demonstrated a decrease in prostate volume in all dogs and no major complications were noted. While this treatment substantially reduced the tumor volume, it did not eradicate it, and additional investigations regarding dose and delivery are necessary. It is unknown how effective DEB-PAE or bland-PAE is when compared to radiotherapy or surgery; however, it does appear to be substantially less morbid without the association of significant genitourinary or gastrointestinal side effects. Additionally, it appears to be effective at reducing prostate and tumor size with varied effects on quality of life and symptoms. While worth consideration, it appears that systemic tumor progression occurs in the face of local tumor control, and further investigation into managing local and distant disease progression continues to be essential. 

Exciting advances in PAE include the development of radioembolization with ^90^Y microspheres (TheraSphere; Boston Scientific; Marlborough, MA, USA). The beta-particle emission from radioactive decay results in a more focused distribution of energy delivery into surrounding tissues, with the majority deposited within 5 mm of the emitting particle. This has many potential advantages regarding the avoidance of acute and late-term radioeffects. A study evaluating the feasibility, safety, and absorbed dose distribution of prostate ^90^Y radioembolization in a canine model of induced BPH was recently completed. Animals were divided into groups based on escalated dose and delivered radioembolic, and dogs served as their own controls as only one prostate lobe underwent treatment. Positron emission tomography/MRI was subsequently performed to evaluate the absorbed dose and volume change. The bladder and rectal wall were exposed to tolerable doses of radiation based on microdosimetry, and a significant volume decrease was noted in all dogs, which correlated positively to an escalated dose. No adverse events were detected in the follow up period. Additionally, there was no non-target tissue damage when tissues were harvested and evaluated microscopically [40]. While seemingly safe with a limited side effect profile, additional research into radioembolization for the treatment of PC is essential. 

Currently, there remains limited published experience with PAE in people and dogs, although it remains a compelling treatment option. Unanswered questions include the advantage of chemotherapy with embolic compared to bland or radioembolization, as well as long-term outcomes compared to other treatment modalities (surgery, radiotherapy) in dogs. The pairing of embolization with concurrent therapies such as IA or IV chemotherapy or external beam radiation and the efficacy of repeat embolization is unknown. While these questions require additional effort to be answered, the clinical role of PAE in dogs with naturally occurring PC is justified and, while technically challenging, appears to be feasible and minimally invasive with a very low rate of procedure-associated complications. 

### 4.2. Intra-Arterial Chemotherapy

There has been some published experience on the utility of chemotherapy with canine PC. Intra-arterial chemotherapy is of notable interest for PC due to the increased drug concentration within the tumor following intra-arterial administration, while also sparing systemic exposure and reducing adverse events (Figure 1). There has been increased focus on bladder cancer and intra-arterial chemotherapy. An early investigation in a rabbit model of bladder cancer evaluated outcomes following IA or IV infusion once a week for three weeks of carboplatin and pirarubicin. All bladder tumors in the IA group decreased in size or disappeared entirely [41]. In people, there appears to be some evidence that IA infusions of cisplatin may decrease bulky tumors and improve outcomes in patients without metastasis for bladder carcinoma [42]. Pairing IA chemotherapy and radiation concurrently as a primary treatment or in the neoadjuvant setting also demonstrated success in improving rates of response while minimizing systemic toxicity for bladder carcinoma [43,44]. Previously, IA chemotherapy (cisplatin) paired with radiation therapy for the treatment of urinary bladder carcinoma in two dogs led to a reduction in tumor size in both and was well tolerated [45]. While encouraging, there is a lack of randomized controlled trials to establish these treatments in canine lower urinary tract carcinomas. In a more recent retrospective study [46], intra-arterial chemotherapy alone was compared to intravenous chemotherapy regarding local short-term effects against spontaneously forming lower urinary tract tumors in dogs. Dogs with prostatic or urothelial carcinomas who received IV or IA carboplatin and an NSAID were included. Ultrasonographic appearance of the tumor prior to and following two doses of IA chemotherapy revealed a significant change in the longest unidimensional measurement, which was not noted in the IV chemotherapy group. While this was retrospectively performed, it is suggestive of some increased benefit to the use of the super-selected delivery of chemotherapy into lower urinary tract carcinomas. An additional prospective study performed to evaluate serum concentration of chemotherapeutics evaluated the IA and IV treatment of lower urinary tract tumors with mitoxantrone, doxorubicin, or carboplatin. The area under curve (AUC) for the serum drug concentration-time was significantly lower after IA mitoxantrone compared to IV, while peak serum concentrations of IA carboplatin were significantly lower compared to equivocal IV and AUC values. Doxorubicin-delivered IA or IV did not demonstrate measurable differences in AUC or peak serum concentrations. While these findings appear mixed, the heterogenous population of treated tumors and patients as well as various factors impacting tumor uptake of chemotherapy were not controlled or specifically evaluated for, and additional controlled studies may be helpful [47]. Ultimately, there appears to be some evidence that IA carboplatin has low systemic exposure and may be effective against lower urinary tract tumors including PC in dogs. This treatment should be considered as a safe and technically feasible treatment with some evidence of increased effectiveness against a resilient tumor.

### 4.3. Palliative Stenting for Urethral and Ureteral Obstruction

#### 4.3.1. Urethral Stenting

The local progression of PC can lead to urethral or ureteral obstruction, for which urethral or ureteral stenting is possible. The transurethral placement of permanent urethral stents to treat malignant urethral obstruction in dogs has been described [48]. Balloon expanded metallic stents or self-expanding metallic stent placement was described, although self-expanding stents may be preferred and are almost exclusively used at the authors’ institutions. The procedure is minimally invasive, and the procedural time tends to be short. In the original report of urethral stents for malignant obstruction, death was not related to urethral obstruction in all dogs for which stents were placed and nine of twelve dogs were continent or mildly incontinent after stent placement, while the remaining three were severely incontinent (2) or had an atonic bladder (1) in one study. Major complications included stent dislodgement in one dog, although none of the dogs had reported tumor in-growth [48]. A second study retrospectively evaluating a larger population of dogs undergoing palliative urethral stent placement showed a high rate of technical success, although 26% of all dogs (5/19 females and 6/23 males) were severely incontinent following stent placement [49]. Interestingly, stent length, diameter, and location were not associated with incontinence or stranguria. Ultimately, 95% of dogs were euthanized following stent placement for reasons unrelated to urethral obstruction [49]. Clinically, urethral stents are an excellent option for resolving the life-threatening condition of obstructive neoplasia in the bladder, urethra, and/or prostate. While urinary incontinence is a risk of this procedure, it appears to severely affect a minority of patients. 

Temporary stents are placed less frequently and typically as a bridging therapy until permanent stents can be placed. These are often rubber or polyurethane and can be temporarily managed by an owner at home [50]. In one study, temporary urethral stents placed for benign or malignant etiologies were successfully placed and well tolerated but led to urinary incontinence in all dogs they were placed in, if they spanned from bladder to urethral orifice, and were associated with complications such as bacteriuria and stent migration [51]. While reasonable as a temporary solution, these are generally not considered adequate as a long-term solution. 

#### 4.3.2. Ureteral Stenting

In people, a ureteral stent placement to relieve ureteral obstruction has been utilized for the palliation of urologic malignant disease. Stents are typically preferred over nephrostomy tubes for improved tolerability, although polymeric and metallic stents had mixed success in maintaining patency following placement in patients [52]. While there are various different devices available, metallic stents generally resist external compressive forces better than polymeric stents. Patency at the time of placement appears to be very high [52,53,54], and overall stent failure over 12 months following placement remains tolerable [52,53]. Compared to metallic stents, polymer stents may have comparable patency at 6 months and significantly diminished patency and associated quality of life at 12 months [55]. In contrast to metallic stents, which may be allowed to indwell for longer, polymer stents may be exchanged as frequently as every 3 months, which can be perceived negatively by a patient [55,56]. 

In dogs with ureteral stents that are placed for benign or malignant ureteral obstruction, stent exchange is often not financially possible or planned for. Additionally, the long-term outcome in dogs following stent placement for malignant obstruction is often poor; therefore, prolonged patency may be less important. In dogs, percutaneous antegrade ureteral stent placement is advocated for relief of malignant ureteral obstruction [50]. Due to the decreased visibility or access to the ureterovesicular junctions, these are most often placed percutaneously and antegrade (Figure 2), which is technically challenging and requires substantial experience with ultrasound-guided and fluoroscopic-guided procedures. Reported complications of this procedure include the inability to successfully place stents percutaneously requiring conversion, the migration of a stent, the disruption of the upper urinary tract, or tumor seeding at skin puncture sites [57,58]. While comparably little is known on how the type of stent device may play a role in immediate and long-term patency in dogs and metallic ureteral stents have not been reported in veterinary patients, the long-term indwelling time of metallic stents may align well with the inability to perform stent exchange in veterinary medicine and could be considered as a future point of investigation in dogs. 

## 5. Concluding Remarks

At this time, there remains a lack of robust trials comparing treatment modalities and long-term outcomes in dogs diagnosed with prostate carcinoma. As such, the role of interventional oncology in the treatment of prostate cancer is still being investigated. Importantly, the practitioner must consider each patient independently while weighing disease burden, stage, and external patient and owner factors. IO techniques remain minimally morbid, with percutaneous access (PAE, intra-arterial chemotherapy, ureteral stenting) or natural orifice access (urethral stenting, ureteral stenting), which is hugely advantageous. While long-term data for dogs undergoing prostate artery embolization or intra-arterial chemotherapy is limited, locoregional effectiveness based on tumor volume reduction was documented in both [14,46], which is encouraging. IA chemotherapy may demonstrate improved effectiveness in bulky disease compared to IV administration, although the long-term control of distant disease progression is unknown and there may be a role for both for disease management [19,46]. Additionally, while no data exist comparing embolization to radiotherapy or surgery, the less invasive nature and high tolerability makes embolization an exciting treatment option for the local prostatic disease. Urethral and ureteral stenting can be used in the treatment of progressive obstructive disease within the lower urinary tract which may mitigate life-limiting complications that are frequent to prostate carcinoma locoregional progression [48,49,50,51,58]. Lower urinary tract stenting appears to be well tolerated and is considered a good option in the palliation of this disease. When considering these treatments, owner finances as well as available resources (e.g., fluoroscopy) and procedural experience will play an important role in what can be offered. Additionally, thorough the evaluation of disease burden to identify the appropriateness of treatments is essential and may include ultrasound, computed tomography angiography, cystoscopy, and/or cystourethrography. 

In conclusion, canine PC remains a diagnostic and therapeutic treatment challenge with guarded long-term prognosis. Despite this, the translation of therapies such as stenting for malignant obstruction and prostate artery embolization to clinical canine PC is encouraging. While surgery and radiotherapy remain valid as locoregional therapies, the inability to obtain consistently good long-term outcomes without a moderate risk of side-effects may limit the tolerance of the associated morbidity of some of these treatments. The role of interventional oncology in the palliative setting (percutaneous or transurethral stenting procedures) or in the direct treatment of PC in dogs (embolization, intra-arterial chemotherapy) is an exciting field that deserves ongoing focus in the future. While ongoing research is needed, there are encouraging therapeutic developments that may allow the optimization of the treatment of PC in dogs and people. 

## Figures and Tables

**Figure 1 vetsci-11-00169-f001:**
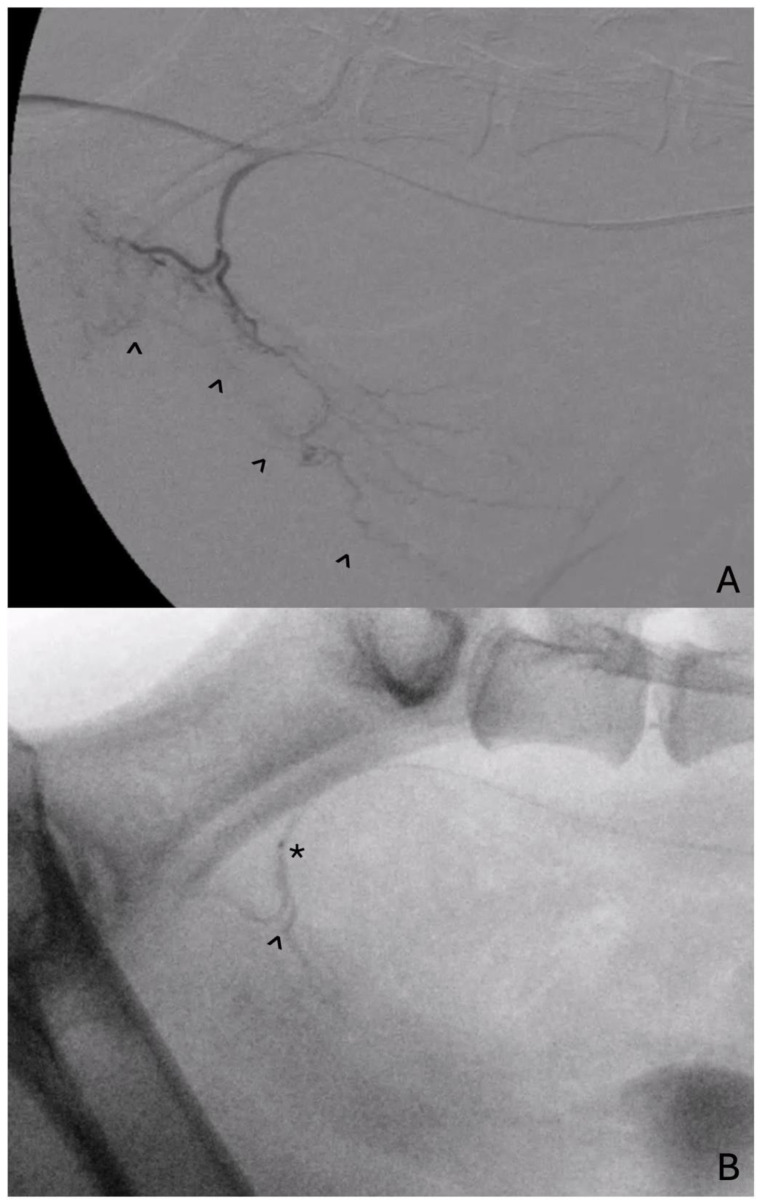
(**A**). Lateral digital subtraction angiogram at the level of the prostatic artery demonstrating extensive neovascularization of the prostate, urethra, and trigone (carat) in a dog with progressive prostatic carcinoma. (**B**). Lateral fluoroscopic view of the same dog during administration of intra-arterial chemotherapy, admixed with contrast (carat), into the prostatic artery via microcatheter (asterisk).

**Figure 2 vetsci-11-00169-f002:**
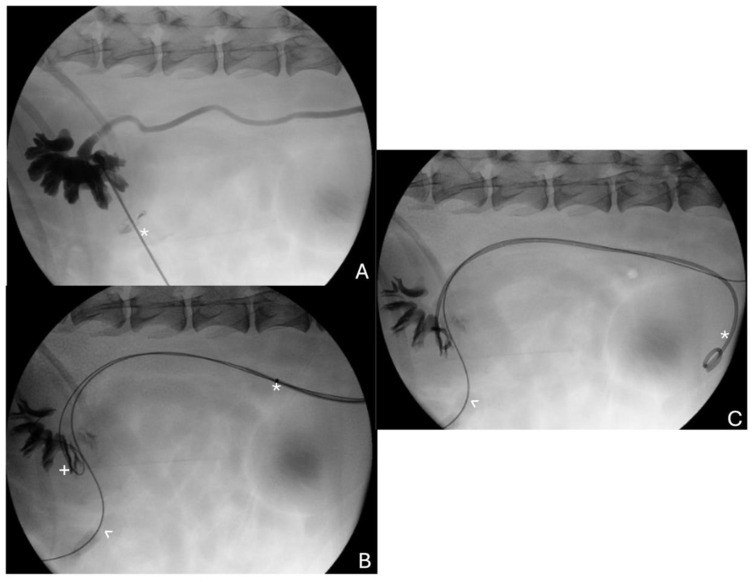
(**A**). Fluoroscopic image of percutaneous renal pelvic access (obliqued, head left) with a needle (asterisk) for diagnostic pyelogram and for antegrade placement of a wire spanning into the bladder and exiting the urethral orifice for fluoroscopic guided percutaneous ureteral stent placement for obstructive urothelial carcinoma. (**B**). Fluoroscopic image of through-and-through wire (carat), long access sheath (asterisk), and second wire (plus) coiled within the renal pelvis that spanned the site of ureterovesicular junction obstruction. (**C**). Fluoroscopic image of ureteral stent (asterisk) in place alongside through-and-through wire (carat) spanning the urinary tract.

## Data Availability

No new data were created or analyzed in this study. Data sharing is not applicable to this article.

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
