# Peer review of "Canine Prostate Cancer: Current Treatments and the Role of Interventional Oncology"

_vetsci, 2024, doi:10.3390/vetsci11040169_

Round 1

Reviewer 1 Report

Comments and Suggestions for Authors

In this paper, the authors present a review of treatments available for canine prostatic tumors. They did not perform a systematic review or meta-analysis, and consequently, the structure of the manuscript is adequate. The language is clear, even though from time to time some sentences could be simplified. Some suggestions may be found in the commented copy of the MS attached to this report.

Regarding the content, some issues need the attention of the authors:

1. A problem common to all the MS sections respect the position of the referenced sources towards the punctuation. it should be placed before punctuation marks, not after.

2. in the abstract - the focus of the MS is the canine prostatic tumors, not men PC; so why spend half the abstract talking about prostate tumors in men? Please revise the abstract so it covers the topic of the paper.

3. Introduction - the first sentence states that prostatic tumors in dogs are more frequent in castrated males. However, this prevalence only applies to prostatic carcinomas, not to those of urothelial origin. Therefore, it would be better to first present the types/variations of prostatic tumors in dogs before reporting the prevalence of the disease.

4. When reporting the outcomes of radiation treatments (as for any other treatments tested in dogs) it would be crucial to understand the effectiveness of the therapeutic approach to have additional information on the treated animals would be presented. That is, all animals were free of visible metastases? for how long were they followed to assess the effect of treatment? Was the type of tumor considered and affected the outcome? these parameters may determine the treatment outcome and the survival or the time free of disease

5. lines 143 -145 - the sentence needs clarification - particularly regarding the fact that this study used dogs with lower urinary tract tumors; was the type of tumors used to group animals? or to analyze the treatment outcome? Do we have results more specific for the prostatic tumors? Which type of PC did the authors identify in this study? It may happen that it was restricted to urothelial carcinoma, as it induced urethral constriction.

6. Briefly describe the toxicity levels for better comprehension. The paper is also interesting for less specialized people and students; so, it should be clear to all of them

7.  Lines 206 -209 - the description presented here points to air embolism, not bland embolization. Do you confirm?
If it actually refers to a study about bland embolization, then it would be advisable to identify the particles used in the experiments.

8. In the section describing radio embolization, the animals used in the experiments had BPH, not prostatic tumors. So, a comment highlighting the fact that this approach was never tested in PC would be important, because the reduction in size does not equal to tumor destruction.

9. In comparison with previous subsections, in the sub-section about different embolization approaches I miss the information regarding the survival time or factual information about the evolution of the condition. Can you add it?

As previously said, the commented copy of the manuscript attached has additional minor suggestions and comments.

Author Response

The authors thank you for the valuable feedback. Specific responses have been described below.

Regarding the content, some issues need the attention of the authors:

1.A problem common to all the MS sections respect the position of the referenced sources towards the punctuation. it should be placed before punctuation marks, not after.

This has been addressed.

  1. in the abstract - the focus of the MS is the canine prostatic tumors, not men PC; so why spend half the abstract talking about prostate tumors in men? Please revise the abstract so it covers the topic of the paper.

Thank you for this comment, the abstract has been amended in lines 16-25:

Abstract: Prostate carcinoma is one of the most common cancers worldwide in men, with over 3 million men currently living with prostate carcinoma[1–3]. In men, routine screening and successful treatment schemes including radiation, prostatictectomy, or hormone therapy has allowed for high survivability[2–4]. Dogs are recognized as one of the only mammals to develop spontaneous development of prostate neoplasia and are an important translational model. Within veterinary medicine, treatment options have historically been limited in efficacy or paired with high morbidity. Recently, less invasive treatment modalities have been investigated in dogs and people and have demonstrated promise. Below, current treatment options available in dogs and people will be reviewed, as well as a discussion of current and future trends within interventional treatment for canine PC.

  1. Introduction - the first sentence states that prostatic tumors in dogs are more frequent in castrated males. However, this prevalence only applies to prostatic carcinomas, not to those of urothelial origin. Therefore, it would be better to first present the types/variations of prostatic tumors in dogs before reporting the prevalence of the disease.

The order of the paragraphs has been altered such that histopathology is discussed initially, to ensure optimal understanding of disease occurrence and susceptibility.

  1. When reporting the outcomes of radiation treatments (as for any other treatments tested in dogs) it would be crucial to understand the effectiveness of the therapeutic approach to have additional information on the treated animals would be presented. That is, all animals were free of visible metastases? for how long were they followed to assess the effect of treatment? Was the type of tumor considered and affected the outcome? these parameters may determine the treatment outcome and the survival or the time free of disease

Thank you for this comment. The studies that are cited are retrospective studies so tumor response to treatment is variably reported, and treatment groups are heterogenous, but certainly we acknowledge the importance of improving the reader’s understanding of outcomes following radiotherapy. As these were retrospective in nature, these studies variably evaluated patient factors within the umbrella of lower urinary tract carcinomas, such as dogs with prostate involvement, which is already specified in the manuscript (like 144-145, 153-154, and line 162-163)

Lines 138-142, 151-153, 161-162 have been added to help expand upon treatment outcomes in dogs undergoing radiotherapy:

All dogs were retrospectively categorized into 3 treatment groups including dogs undergoing first line concurrent chemoradiotherapy (1), first-line chemotherapy >1 mo prior to initiating radiotherapy who did not have evidence of tumor progression (2), and dogs receiving radiotherapy as a salvage following locoregional failure (3).

In all dogs, median time to local progression was 343 days and was reported in 59% of dogs. Locoregional failure rates per group was 56%, 50%, and 75% for groups 1, 2, and 3

 Of the owners who completed standardized post-treatment questionaries, 60% perceived improved quality of life while 30% reported unchanged

  1. lines 143 -145 - the sentence needs clarification - particularly regarding the fact that this study used dogs with lower urinary tract tumors; was the type of tumors used to group animals? or to analyze the treatment outcome? Do we have results more specific for the prostatic tumors? Which type of PC did the authors identify in this study? It may happen that it was restricted to urothelial carcinoma, as it induced urethral constriction.

Unfortunately, due to the retrospective nature of the cited paper, most of this information is not available but how the animals were categorized for analysis is clarified in line 138-142, and additional information relative to tumor type/anatomic involvement is listed in line 142-144:

All dogs were retrospectively categorized into 3 treatment groups including dogs undergoing first line concurrent chemoradiotherapy (1), first-line chemotherapy >1 mo prior to initiating radiotherapy who did not have evidence of tumor progression (2), and dogs receiving radiotherapy as a salvage following locoregional failure (3). Fifty-one dogs with primary genitourinary urothelial carcinoma were included, and further categorized into bladder (19), prostate (17), urethral (4), and 11 that were multifocal within the urinary tract.

  1. Briefly describe the toxicity levels for better comprehension. The paper is also interesting for less specialized people and students; so, it should be clear to all of them

Citation (Ladue, T.; Klein, M.K.; Veterinary Radiation Therapy Oncology Group Toxicity Criteria of the Veterinary Radiation Therapy Oncology Group. Vet. Radiol. Ultrasound Off. J. Am. Coll. Vet. Radiol. Int. Vet. Radiol. Assoc. 2001, 42, 475–476, doi:10.1111/j.1740-8261.2001.tb00973.x.) has been added for the readers’ reference in understand toxicity grades

  1. Lines 206 -209 - the description presented here points to air embolism, not bland embolization. Do you confirm?If it actually refers to a study about bland embolization, then it would be advisable to identify the particles used in the experiments.

It does not point to air embolism, but embolization with bead embolic. This has been clarified in line 211:

Prostate artery embolization with embolic beads in dogs with spontaneous PC has also been evaluated

  1. In the section describing radio embolization, the animals used in the experiments had BPH, not prostatic tumors. So, a comment highlighting the fact that this approach was never tested in PC would be important, because the reduction in size does not equal to tumor destruction.

Thank you for this comment. Line 289-290 has been added to clarify:

While seemingly safe with limited side effect profile, additional research into radioembolization for treatment of PC is essential.

  1. In comparison with previous subsections, in the sub-section about different embolization approaches I miss the information regarding the survival time or factual information about the evolution of the condition. Can you add it?

Thank you for this comment. The literature cited in this section does not have long-term survival data, and evolution of the condition is reported in lines 259-261 for bland-PAE, lines 262—266 for DEB-PAE, and lines 284-286 for radioembolization.

Reviewer 2 Report

Comments and Suggestions for Authors

The article provides a thorough overview of prostate carcinoma in dogs, covering various aspects including epidemiology, diagnosis, and treatment options. This comprehensive approach ensures that readers gain a well-rounded understanding of the topic. Authors discusses technical aspects of IO procedures in detail, such as the anatomy of the prostatic vasculature, selection of embolic materials, various techniques such as prostate artery embolization (PAE), intraarterial chemotherapy, and palliative stenting for urethral and ureteral obstruction and potential complications associated with stent placement. This information provides valuable guidance for veterinary professionals performing these procedures. Authors integrates findings from both human and veterinary studies to support its discussion on the efficacy and safety of IO approaches in canine prostate carcinoma. The article is well-structured, with distinct sections for introduction, diagnosis, and current treatments. This organization facilitates readability and helps readers navigate through the content efficiently. The article acknowledges limitations and challenges in the diagnosis and treatment of canine prostate carcinoma, such as the lack of screening tools and the risks associated with certain diagnostic and therapeutic approaches. This transparency adds depth to the discussion and provides a realistic perspective on the topic. Authors emphasizes clinical relevance by discussing practical aspects such as diagnostic methods, treatment modalities, clinical relevance of IO techniques in veterinary oncology, highlighting their potential benefits in improving quality of life and reducing morbidity for dogs with prostate carcinoma. This focus on real-world implications enhances the article's utility for veterinary professionals dealing with canine prostate carcinoma.

.

From the reviewer's point of view, it is also necessary to indicate the points of the article that can be improved;

Visual aids such as figures, tables, or diagrams could enhance the presentation of complex information, such as diagnostic algorithms or treatment outcomes. Incorporating visual elements would not only improve the clarity of the article but also engage readers more effectively.  While the article discusses various treatment modalities, it does not offer a comparative analysis of their efficacy, safety profiles, or cost-effectiveness. Providing a comparative overview could assist readers in making informed decisions regarding the selection of treatment options based on factors such as patient characteristics and available resources. The article lacks comparative analysis of the efficacy and safety of different IO approaches compared to traditional treatment modalities such as surgery and radiotherapy. Including comparative data would help readers understand the relative merits of IO techniques in the context of existing treatment options. While authors discuss short-term outcomes and technical feasibility of IO procedures, it provides limited insight into long-term treatment outcomes and recurrence rates in dogs undergoing these interventions. Including data on long-term follow-up would enhance the article's discussion on treatment efficacy and durability. Authors does not extensively discuss criteria for patient selection or factors influencing treatment decisions regarding the choice between different IO approaches. Including guidance on patient selection criteria would assist veterinary practitioners in making informed decisions tailored to individual patient characteristics and disease stage. The article does not address cost considerations associated with IO approaches, such as the financial implications of embolization materials or stent placement procedures. Including a discussion on cost-effectiveness would provide valuable insight for veterinarians and pet owners considering these treatment options.

Author Response

Visual aids such as figures, tables, or diagrams could enhance the presentation of complex information, such as diagnostic algorithms or treatment outcomes. Incorporating visual elements would not only improve the clarity of the article but also engage readers more effectively.  While the article discusses various treatment modalities, it does not offer a comparative analysis of their efficacy, safety profiles, or cost-effectiveness. Providing a comparative overview could assist readers in making informed decisions regarding the selection of treatment options based on factors such as patient characteristics and available resources. The article lacks comparative analysis of the efficacy and safety of different IO approaches compared to traditional treatment modalities such as surgery and radiotherapy. Including comparative data would help readers understand the relative merits of IO techniques in the context of existing treatment options. While authors discuss short-term outcomes and technical feasibility of IO procedures, it provides limited insight into long-term treatment outcomes and recurrence rates in dogs undergoing these interventions. Including data on long-term follow-up would enhance the article's discussion on treatment efficacy and durability. Authors does not extensively discuss criteria for patient selection or factors influencing treatment decisions regarding the choice between different IO approaches. Including guidance on patient selection criteria would assist veterinary practitioners in making informed decisions tailored to individual patient characteristics and disease stage. The article does not address cost considerations associated with IO approaches, such as the financial implications of embolization materials or stent placement procedures. Including a discussion on cost-effectiveness would provide valuable insight for veterinarians and pet owners considering these treatment options.

We thank the reviewer for this feedback and appreciate the need for further guidance. While long-term data does exist for some treatment modalities, it is not available in all. Additionally, there remain no strong RCT that compares treatment modalities such as surgery to RT to embolization, and the authors are reluctant to summarize available data too simply such that assumptions are made on grounds that have not been justified in the literature. That being said, we acknowledge that providing better direction may be helpful, and a paragraph has been added at the end of the manuscript:

At this time, there remains a lack of robust trials comparing treatment modalities and long-term outcomes in dogs diagnosed with prostate carcinoma. As such, the role of interventional oncology in the treatment of prostate cancer is still being investigated. Importantly, the practitioner must consider each patient independently while weighing disease burden, stage, and external patient and owner factors. IO techniques remain minimally morbid, with percutaneous access (PAE, intra-arterial chemotherapy, ureteral stenting) or natural orifice access (urethral stenting, ureteral stenting), which is advantageous. While long-term outcome data for dogs undergoing prostate artery embolization or intra-arterial chemotherapy is limited, locoregional effectiveness based on tumor volume reduction has been documented in both[15,47], which is encouraging. IA chemotherapy may demonstrate improved effectiveness in bulky disease compared to IV administration, although long term control of distant disease progression is unknown and there may be a role for both for disease management[20,47]. Additionally, while no data exists comparing embolization to radiotherapy or surgery, the less invasive nature and high tolerability makes embolization an exciting treatment option for the local prostatic disease. Urethral and ureteral stenting can be used in the treatment of progressive obstructive disease within the lower urinary tract which may mitigate life-limiting complications that are frequent to prostate carcinoma locoregional progression[49–52,59].  Lower urinary tract stenting appears to be well tolerated and is considered a good option in the palliation of this disease. When considering these treatments, owner finances as well as a available resources (eg. fluoroscopy), and procedural experience will play an important role in what can be offered. Additionally, thorough evaluation of disease burden to identify appropriateness of treatments is essential and may include ultrasound, computed tomography angiography, cystoscopy, and/or cystourethrography.

Additionally, 2 figures have been added to enhance readability. 

Line 296-299: 

Figure 1. A. Lateral digital subtraction angiogram at the level of the prostatic artery demonstrating extensive neovascularization of the prostate, urethra and trigone (carat) in a dog with progressive prostatic carcinoma. B. Lateral fluoroscopic view of the same dog during administration of intra-arterial chemotherapy, admixed with contrast (carat), into the prostatic artery via microcatheter (asterisk).

Line 362-367: 

Figure 2: A. Fluoroscopic image of percutaneous renal pelvic access (obliqued, head left) with a needle (asterisk) for diagnostic pyelogram and for antegrade placement of a wire spanning into the bladder and exiting the urethral orifice for fluoroscopic guided percutaneous ureteral stent placement for obstructive urothelial carcinoma. B. Fluoroscopic image of through-and-through wire (carat), long access sheath (asterisk), and second wire (plus) coiled within the renal pelvis that have spanned the site of ureterovesicular junction obstruction. C. Fluoroscopic image of ureteral stent (asterisk) in place along side through-and-through wire (carat) spanning the urinary tract.

Reviewer 3 Report

Comments and Suggestions for Authors

The review is well done, although it adds nothing to the current knowledge on prostate cancer present in the literature, the presentation of all the data and their analysis makes the paper very useful for the clinician and for the development of new studies.

Line 45: specify the meaning of the acronym “UC” Line 86: specify the meaning of the acronym "MST"

Line 104: specify the meaning of the acronym “RCTs”

Line 118-123: I would also point out the fact, in addition to incontinence, that it is a highly destructive surgery, with opening of the pelvis and that animal welfare must therefore also be considered.

For some authors (Kutzler 2022 for example) prostatectomy should be considered malpractice

 Line 146: specify the meaning of the acronym “EFS and OST” ù

Line 154: Acronyms must be explained the first time they are used

Line249,251,266:replace intraarterial whit “intra-arterial”

Line 336: specify the meaning of the acronym “UVJs

Author Response

Thank you for this feedback, below is a summary of the proposed edits and the responses of the authors.

Line 45: specify the meaning of the acronym “UC” Line 86: specify the meaning of the acronym "MST"

The acronym UC has been clarified in line 30, and MST has been clarified in line 83

Line 104: specify the meaning of the acronym “RCTs”

This has been clarified in line 101

Line 118-123: I would also point out the fact, in addition to incontinence, that it is a highly destructive surgery, with opening of the pelvis and that animal welfare must therefore also be considered.

For some authors (Kutzler 2022 for example) prostatectomy should be considered malpractice

Line 117-119 has been rephrased to emphasize the morbidity of the procedure. The authors do not consider that the primary literature available evaluating prostatectomy in dogs suggests that this procedure can be labeled malpractice. The authors agree that emphasizing the morbidity of this procedure and the need to seek alternatives is important.

While surgery is a viable option, it remains moderately morbid and reasonably complicated, with a high risk of urinary incontinence. Therefore, a need for alternative loco-regional therapies is clear.

 Line 146: specify the meaning of the acronym “EFS and OST” ù

This has been clarified in lines 138

Line 154: Acronyms must be explained the first time they are used

This has been addressed in line 155

Line249,251,266:replace intraarterial whit “intra-arterial”

This is done

Line 336: specify the meaning of the acronym “UVJs”

This is done

Round 2

Reviewer 1 Report

Comments and Suggestions for Authors

In the revised manuscript, The authors responded in full to my questions.

Therefore, I recomment this paper for publication in Veterinary Sciences